# Mechanically Enhanced Nanocrystalline Cellulose/Reduced Graphene Oxide/Polyethylene Glycol Electrically Conductive Composite Film

**DOI:** 10.3390/nano12244371

**Published:** 2022-12-08

**Authors:** Pengbo Xie, Ying Ge, Yida Wang, Jing Zhou, Yuanyuan Miao, Zhenbo Liu

**Affiliations:** Key Laboratory of Bio-Based Material Science and Technology of Ministry of Education, Northeast Forestry University, Harbin 150040, China

**Keywords:** nanocellulose, graphene, polyethylene glycol, composite film, flexible conductive material

## Abstract

Traditional conductive materials do not meet the increasing requirements of electronic products because of such materials’ high rigidity, poor flexibility, and slow biodegradation after disposal. Preparing flexible conductive materials with excellent mechanical properties is an active area of research. The key to flexible conductive materials lies in the combination of the polymer matrix and conductive components. This combination can be achieved by making a film of renewable nano-microcrystalline cellulose (NCC) and reduced graphene oxide (rGO) with excellent electrical conductivity—by simple filtration and introducing polyethylene glycol (PEG) to enhance the functionality of the composite film. Graphene imparted conductivity to the composite film, which reached 5.67 S·m^−1^. A reinforced NCC/rGO/PEG-4 composite film with a thickness of only 21 μm exhibited a tensile strength of 30.56 MPa, which was 83% higher than that of the sample without PEG (16.71 MPa), and toughness of 727.18 kJ·m^−3^, which was about 132% higher than that of the control sample (NCC/rGO, 313.86 kJ·m^−3^). This ultra-thin conductive composite film—which can be prepared simply, consists of environmentally sustainable and biodegradable raw materials, and exhibits excellent mechanical properties—has substantial potential for applications in e.g., flexible electronic wearable devices, electrodes, and capacitors.

## 1. Introduction

With the development of science and technology, electronic products tend to be increasingly miniaturized and lightweight. Because of their high rigidity, traditional metallic conductive materials do not meet such requirements; which is especially pertinent to e.g., touch displays, solar panels, strain sensors, stretchable electrodes, and electronic skin [1,2,3,4,5]. Therefore, preparation of flexible conductive materials—which are indispensable for electronic devices—is an active area of research globally. The core issues in preparing flexible conductive materials are conductivity and flexibility, and the key lies in selecting conductive components and flexible polymer substrate [6].

Cellulose is a natural polymer material that can be obtained from wood, palm, straw, cotton, and linen; and exhibits good biocompatibility, regenerability, and biodegradability [7,8,9,10,11,12]. Cellulose sugar chains can be stacked into a tight lamellar structure by intermolecular hydrogen bonding, which imparts good support and protection when used as a membrane substrate material. Nanocellulose refers to cellulose with at least one dimensional space size within the nanoscale (1–100 nm); can be prepared chemically or mechanically from natural cellulose; and exhibits properties such as a high aspect ratio, large specific surface area, high hydrophilicity, high strength, and low coefficient of thermal expansion [13]. Bertsch found that NCC can form stable films at the air/water interface because of their amphiphilic nature [14]. The cellulose nanofibrils/nano-microcrystalline cellulose (CNF/NCC) films prepared by simple extraction by Sun et al. exhibited better thermal stability and a higher coefficient of thermal expansion compared with commercial porous polymeric battery diaphragm films (11.86–17.65 ppm/K and 178.90 ppm/K, respectively) [15]. Perdoch reported a carboxylated nanocellulose coating that substantially enhanced the physical properties of paper; especially the tensile index, tear resistance, surface roughness, and bending strength [16]. Nanocellulose not only exhibits the characteristics of macrocellulose but also exhibits the excellent properties of nanomaterials. Nanocellulose is often used as a template carrier or reinforcing unit in composite materials; and is widely used in e.g., mechanical reinforcing, biomedical, adsorbent, catalytic, and packaging materials [17,18,19]. In particular, researchers have made substantial progress in using nanocellulose in flexible electronic substrates for wearable and bioelectronic devices [20].

Nanocellulose does not exhibit electrical conductivity, but can form a continuous conductive network in a film matrix upon introducing a conductive medium. A nanocellulose-based composite film can be prepared that exhibits electrical properties yet maintains the advantages of nanocellulose. Commonly used conductive fillers are metal and carbon-based fillers. Graphene, a two-dimensional nanomaterial, is useful in this context. Its carbon atoms form covalent bonds by sp2 hybridization with three adjacent carbon atoms, forming a hexagonal single-layer honeycomb structure. This structure imparts unique characteristics, rendering it an excellent electrode [21]. Lv developed a low-cost, environmentally sustainable electrode with a specific capacitance of up to 1144.3 F·g^−1^ and good cycle stability by using an independent high-porous nano-mixed aerogel film [composed of carbon fiber/molybdenum disulfide (MoS_2_)/reduced graphene oxide (rGO) as an electrode, and H_2_SO_4_/polyvinyl alcohol (PVA) gel as an electrolyte] [22]. Zhang prepared positively charged graphene oxide (GO) and negatively charged GO sheets into a composite film, assembled them into mixed filaments, and chemically reduced them to obtain conductive filaments with an average tensile strength of 109 ± 8 MPa as well as a conductivity of 3298 ± 167 S·m^−1^ (which provides a new means of understanding the interactions between GO and nanocellulose; and of designing macroscopic, assembled, and functionalized structures) [23]. Kuang constructed sodium lignosulfate (LS)/polyacrylamide (PAM)/GO composite conductive hydrogels with an EC of 0.0317 S·m^−1^: by using sodium LS as well as PAM as substrates, and introducing GO [24]. Incorporating graphene could substantially improve the EC of the composites. Wang used a one-step hydrothermal method to prepare graphene–cellulose aerogels that exhibited the following: high capacitance (specific capacitance of 202 F·g^−1^ at a scan rate of 5 mA/cm^2^), and the polyvinyl alcohol (PVA)/graphene/cellulose nanofibrils ternary aerogel, which can withstand 628 times its own mass and has a thermal conductivity of 0.044 ± 0.005 W/mK at room temperature, has excellent mechanical properties and thermal stability and good mechanical properties, thermally stable performance, ability to withstand 628× their mass, and a thermal conductivity of 0.044 ± 0.005 W/mK at room temperature [25,26]. The electrical properties of graphene can be combined with the mechanical properties of nanocellulose, and the composites can be prepared to meet practical needs by adding a third component for functional reinforcement. However, because direct preparation of monolayer graphene is cumbersome, rGO was used instead in this study. Cellulose-based films are inflexible mainly due to strong hydrogen bonds between cellulose polymer chains, plasticizers such as glycerol and propylene glycol are usually added to improve the stability of nanocellulose films, by breaking the strong hydrogen bonds between the nanocellulose and forming new connections with them [27]. Glycerol increases flexibility yet decreases the thermal stability of the material because of its relatively small molecular weight and the presence of a large number of hydroxyl groups [28]. Therefore, in this study, polyethylene glycol (PEG; a non-toxic, biocompatible, and biodegradable plasticizer) with a smaller hydrophilicity than glycerol was incorporated.

In this work, a method for the preparation of NCC/rGO/PEG composite films was proposed using PEG as a plasticizer. rGO imparts electrical conductivity to the composite film, and the addition of PEG can further enhance the thermal stability and mechanical properties, especially flexibility, of the composite film by forming hydrogen bonding connections with the other two components, respectively. In this way, the potential of the composite film for flexible electronic materials is enhanced.

## 2. Materials and Methods

### 2.1. Materials and Chemicals

Microcrystalline cellulose powder, particle size 50 μm. Graphite crystals, purity >99 wt.%, were obtained from Jiangsu Changjiade High-tech Carbon Materials Co., Ltd. (Changde, China). Polyethylene glycol, average molecular number of 400, was manufactured by Shanghai Shaoyuan Chemical Co., Ltd. (PEG, Shanghai, China). Hydrogen peroxide (H_2_O_2_), concentration 30 wt.%, was from Tianjin Fuyu Fine Chemical Co., Ltd. (Tianjin, China). Sulfuric acid (H_2_SO_4_), concentration 98 wt.%, potassium permanganate powder (KMnO4), and sodium nitrate (NaNO_3_) were provided by Tianjin Comiou Chemical Reagent Co., Ltd. (Tianjin, China). Hydroiodic acid (HI), concentration 58%, sodium hydroxide (NaOH), and hydrogen chloride (HCl) were provided by Shanghai Aladdin Biochemical Technology Co., Ltd. (Shanghai, China). The water used in the experiment was all deionized water.

### 2.2. Preparation of Graphene Oxide (GO)

Preparation of GO by a modified Hummers method [29]. In an ice-water bath at 4 °C, 2 g of graphite powder and 1 g of sodium nitrate were added sequentially to 46 mL of H_2_SO_4_ (98%) and stirred well. The temperature was controlled at 5 °C to 10 °C, then 6 g of KMnO_4_ was slowly added; the reaction was carried out for 90 min. Then the mixture was heated in a water bath at 35 °C to 40 °C for 30 min; the temperature of the mixture increased substantially after adding 92 mL of deionized water dropwise. After the mixture cooled to room temperature, H_2_O_2_ was added dropwise until the color change of the mixture is no longer apparent. The resulting product was ultrasonically dispersed into a suspension for 30 min and was centrifuged with a high-speed centrifuge (speed 1500 r/min for 30 s). The upper solution was undisturbed for 24 h. After the mixture separated into distinct layers, the lower layer of sediment was centrifuged 8× *g* (speed 6000 r/min for 180 s) and dialyzed until pH-neutral. Finally, the GO suspension concentration was determined by the drying method to be 0.4% *w*/*v*.

### 2.3. Preparation of Nano-Microcrystalline Cellulose (NCC)

Preparation of NCC by sulfuric acid method [30]. To 30 mL of 50 *v*/*v*% H_2_SO4, 2 g of microcrystalline cellulose powder was added, heated at 60 °C, and stirred for at least 2 h to complete hydrolysis. Then the hydrolyzed suspension was removed, 500 mL of deionized water was added, and mixture was left to stand for 12 h. The upper clear layer was removed after delamination and the lower suspension was dialyzed in a dialysis bag to pH-neutral. Then the suspension was ultrasonically fibrillated with an ultrasonic cell crusher for 15 min to obtain an NCC suspension, and the concentration was measured.

### 2.4. Preparation of Nanocrystalline Cellulose/Reduced Graphene Oxid/Polyethylene Glycol Composite Films (NCC/rGO/PEG)

The composite films were prepared by vacuum filtration (Figure 1). 10 mL of NCC suspension with a concentration of 0.3% *w*/*v* was measured and 7.5 mL of 0.4% *w*/*v* GO suspension and 5 mL of 0.4% *w*/*v* PEG solution were added to prepare a mixture of NCC:rGO:PEG = 1.5:1.5:1. This process was repeated to prepare 5 sets of mixed solutions with the mass ratio of each substance as shown in Table 1. In an ice bath, the mixture was treated with ultrasound for 5 min and then stirred for 2 h. The mixture was filtered with a circulating water multipurpose vacuum pump, sand core funnel device, and a polytetrafluoroethylene (PTFE) membrane. After the prepared composite was completely peeled off from the PTFE membrane by using acetone, the composite was immersed 10 min in 58 *v*/*v*% hydroiodic acid (HI) for reduction of GO. Finally, the residual HI on the sample was washed off with 45 *v*/*v*% ethanol, and then dried in a blast drying oven. The samples were termed NCC/rGO/PEG-1, NCC/rGO/PEG-2, NCC/rGO/PEG-3, and NCC/rGO/PEG-4; and the NCC/rGO membrane without addition of PEG was used as the control group.

### 2.5. Characterization

The thickness of the composite film samples was measured with a digital micrometer. Field-emission scanning electron microscopy (SEM; Apreo S HiVac) was used to characterize the microscopic morphology of the samples with a scanning acceleration voltage of 12.5 kV, the samples were snap-frozen with liquid nitrogen to make them brittle-broken and then fixed on the sample stage with conductive tape and then sprayed with gold before testing. Fourier-transform infrared (FTIR) spectra were obtained with a micro-infrared spectrometer (Nicolet iN10) with a scan range of 550–4000 cm^−1^, a scan number of 20, and a resolution of 4 cm^−1^. X-ray diffraction (XRD) (Rigaku SmartLab-SE) was performed over a scanning range of 5° to 90° at a scanning speed of 5°/min, a Cu-Kα X-ray light source was used with a tube voltage of 40 kV and a current of 40 mA. The thermal stability was characterized with a thermo-gravimetric analyzer (TG209F1) under a nitrogen atmosphere with a ramp rate of 10 °C/min and a test temperature range of room temperature to 600 °C. Resistivity was determined with a multifunctional digital four-probe tester (ST-2258C), and the correction factor was reset before testing. All samples were cut into squares of 10 mm in length and 10 mm in width, and the results were averaged over all samples. The mechanical properties were measured with a microcomputer-controlled electronic mechanical testing machine (CMT6103 with MTS system) in plastic-film tensile properties mode with a loading rate of 1 mm/min and a scale distance of 5 mm. All samples were cut into strips of 25 mm in length and 10 mm in width before testing, and the test results were the average of all samples.

## 3. Results and Discussion

### 3.1. Scanning Electron Microscopy (SEM)

Cross sections and planes of the samples were observed by SEM (Figure 2): where a, b, c, d, and e are SEM images of the cross sections of various NCC/rGO samples [NCC/rGO, NCC/rGO/PEG-(1–4), respectively]; ai, bi, ci, di and ei show surface SEM images of the samples; and aii, bii, cii, dii and eii show enlarged cross-sectional regions. It can be seen that the surface of the NCC/rGO sample in Figure 2ai is relatively smooth, and the bar shape of NCC can be vaguely seen. The cross-section of the NCC/rGO film in Figure 2a is very flat, and the magnification (Figure 2aii) shows a regular and orderly lamellar structure with many protruding flakes, which is because the connection is very tight, so many large pieces of rGO flakes were pulled out when the sample was made. The surfaces of the samples were not smooth as seen by an unaided eye, and there were irregular warped flakes on the surface, which gradually increased in number with increasing PEG content. Both in the cross sections and local magnifications, the composite film materials exhibited obvious lamellar structures and good interfacial bonding. This is because of the hydrogen bonding between the oxygen-containing groups on the GO surface and the hydrophilic groups in the NCC, which enabled the GO lamellae to penetrate and disperse in the lamellae of the NCC. With increasing PEG content, the cross sections of the samples were no longer flat; cracks and gaps were evident between the closely stacked lamellae. This result is because of the small molecular weight of the PEG and the large number of hydroxyl groups that facilitated interactions with the polymer NCC, and thus filled into the NCC–rGO lamellae [31]. In Figure 2dii, it can seen that there are some reticular connections between the sample NCC/rGO/PEG-3 lamellae, and the increase of PEG content promotes the formation of intermolecular hydrogen bonding connections between PEG and NCC and GO to each other, which in turn enhances the mechanical properties. The marked place in Figure 2eii is the accumulation produced by PEG, and the composite film will obviously show the characteristics of PEG when the content of PEG is higher. This conclusion is also verified by the mechanical property test results that NCC/rGO/PEG-4 has the maximum toughness. Thus, the quantity of added PEG substantially impacted the mechanical properties of the composite film.

### 3.2. Fourier Transform Infrared (FTIR)

Figure 3 shows FTIR spectra of the GO, NCC, and NCC/rGO/PEG composite films. The GO films exhibited obvious characteristic peaks of oxygen-containing groups: stretching vibration (–OH) at 3221 cm^−1^, stretching vibration at 1737 cm^−1^, in-plane bending vibration (–OH) at 1394 cm^−1^, and vibrational absorption (C–O–C) at 1073 cm^−1^. In addition, the characteristic peak at 1631 cm^−1^ corresponds to the carbon-carbon skeleton of GO (C=C). These functional groups demonstrate the preparation of GO. The infrared spectrogram of NCC exhibited a strong absorption peak at 3309 cm^−1^, which corresponds to the telescopic vibrational absorption of the hydroxyl group (–OH); the telescopic and bending vibrational peaks of C–H were evident at 2899 and 1427 cm^−1^. A weak absorption peak was at 1643 cm^−1^, corresponding to the stretching vibration of C=C. The peak at 1311 cm^−1^ corresponds to the bending vibration of –OH. The peaks at 1023 and 1106 cm^−1^ correspond to the C–O–C stretching vibration of cellulosic alcohol and the stretching vibration of the C–C backbone, respectively. The 896 cm^−1^ –OH stretching vibration absorption peak is characteristic of the β-glycosidic bond between cellulose-dehydrated glucose units [32]. Thus, the chemical structure of NCC prepared with concentrated sulfuric acid was not disrupted and retained the basic chemical structure of cellulose [33].

The FTIR curves of rGO film and NCC/rGO/PEG composite films indicate that the absorption intensity of each oxygen-containing group (–OH, C=O, C–OH, and C–O–C) in GO was weakened to a large extent after reduction with HI. The characteristic peaks of the C–OH and C=O groups were almost no longer evident, and the peak intensities of the C–O–C and –OH groups decreased substantially, which indicates that the reduction was good. After adding PEG and rGO, the wave numbers of most functional groups were almost not shifted compared with the FTIR curves of NCC, the O-H stretching vibration peak at 3309 cm^−1^, the C-H stretching vibration and bending vibration peaks at 2899 cm^−1^ and 1427 cm^−1^, and the C-O-C stretching vibration peak at 1023 cm^−1^ increase in peak intensity with increasing PEG content. However, the characteristic broad band of the –OH group at 3309 cm^−1^ was substantially weaker than that of NCC in terms of peak intensity, indicating that the hydroxyl stretching vibration in NCC was limited—which might be because the hydrogen bonds enabled by the hydroxyl group in NCC were influenced by the C–O–C groups in rGO and PEG, yet this influence was gradually weakened with decreasing rGO content. The composite film featured the characteristic peak of the ester group at 1720 cm^−1^ for two reasons: first, the carboxyl group on the surface of NCC might have undergone esterification with the hydroxyl group of PEG; second, a small quantity of incompletely reduced C=O in rGO caused the emergence of ester group characteristic peaks [34]. The C–O–C stretching vibration at 1023 cm^−1^ was strongly limited by the ester group and the intensity of the absorption peak was substantially weaker compared with that of NCC [35].

### 3.3. X-ray Diffraction (XRD)

Regarding XRD of the composite film (Figure 4), the sample GO has a sharp diffraction peak at 2*θ* = 11.7°, which corresponds to the crystalline surface of graphite oxide (001), and the intensity of the composite film at this peak position is greatly reduced after reduction, indicating that GO is successfully reduced in the composite film, but a small amount of incompletely reduced GO is still present [36]. This is consistent with the conclusion of the FTIR tests that the oxygen-containing functional groups were substantially reduced but still present in small quantities after reduction, indicating that the chemical method afforded incomplete reduction of GO. In accordance with the XRD spectrum of the NCC/rGO/PEG composite film, the diffraction peak at 2*θ* = 11.7° was gradually no longer evident with increasing PEG and decreasing rGO. The characteristic peak of PEG was evident at 2*θ* = 15.5°, with a slight shift to decreasing value of the peak position compared with its typical characteristic peak. In accordance with Bragg’s formula, this implies that the crystalline spacing was reduced, indicating that PEG was completely embedded within the composite film matrix.

GO removes the oxygen-containing functional groups during the reduction, which increases the graphene sheet spacing. However, as the graphene content decreases, insertion of NCC and PEG between the graphene sheets disrupts the π–π stacking of the graphene sheets. Afterward, the graphene sheets rejoin tightly by π–π bonds—which explains the enhanced mechanical properties of the composite films. The peak at 2*θ* = 23.4° is characteristic of the nanocellulose (002) crystalline surface. The intensity of the peak gradually decreased with the increase in PEG content, which indicates formation of hydrogen bonds between PEG and the membrane matrix. This interpretation is consistent with the fact that an increasing content of nanocellulose corresponds to an increasing intensity of the diffraction peak [37,38]. In the XRD pattern of the composite film, no characteristic peak of rGO was evident, but a diffraction peak was evident at 2*θ* = 48.9° (indicating that adding rGO did not change the crystal structure of NCC; rGO was uniformly dispersed in the composite film matrix; and rGO interacted with NCC in a manner that formed a regular, ordered stacking structure) [39,40,41]. The diffraction peak at 48.9° was slightly broadened and weakened after adding PEG (indicating that the original tight stacking of the lamellar structure was disrupted with the change of the ratio of each component of the composite film) and the components re-formed the original three-dimensional structure (which is also consistent with the SEM results [42].

### 3.4. Thermogravimetric (TG)

Figure 5 and Figure 6 show thermal weight loss curves [thermogravimetric (TG)] and first-order differential thermal weight loss curves [differential thermogravimetric (DTG)] of NCC as well as the corresponding composite film; there are clear differences. The weight loss of NCC was mainly divided into two stages: low- and high-temperature regions. As the temperature increased to 50 °C, the thermal weight loss curve began to exhibit a slow downward trend, which mainly corresponds to the evaporation of water. The high-temperature weight loss stage was in the range of 200 °C to 330 °C, and the TG curve exhibited a rapid drop, which corresponds to decomposition and oxidative degradation of NCC. In this region (the largest thermal weight loss zone), the total mass loss was ca. 54.5%, and the maximum thermal weight loss temperature was 309 °C, align with the literature [43] that the maximum thermal weight loss of nanocellulose paper is 327 °C. The oxidative degradation of carbon matter was from 330 °C to 600 °C, in which the weight loss rate was more moderate; and the thermal weight loss of the samples in this stage was ca. 13.5%. After adding rGO and PEG, the composite films exhibited the same weight loss rate (7%) as NCC at the initial weight loss stage, except for NCC/rGO, which exhibited a weight loss rate of 11%. The maximum rate of thermal degradation of the composite films was from 190 °C to 200 °C; the maximum thermal degradation temperature of 200 °C was the highest for NCC/rGO/PEG-1, and its mass loss rate was the smallest (49%), which was ca. 8% lower than that of the sample with the largest loss rate (NCC/rGO). Thus, adding PEG can improve the thermal stability of the composite film; further verifying the conclusion from SEM that there were substantial interactions between NCC and PEG, and the corresponding decreased mass loss of the composite film. However, with increasing content of plasticizer PEG in the composite film, the TG decreases because the relative molecular mass of PEG is smaller so molecular movement is easier. Therefore, the thermal stability of the composite films was improved after adding PEG; NCC/rGO/PEG-1 exhibited the best thermal stability of the tested composites.

### 3.5. Electrical Conductivity

After reducing GO, the rate of electron transfer is affected because of factors such as incomplete removal of oxygen-containing functional groups and incomplete recovery of the structural lattice defects generated during the oxidation [44]. Therefore, rGO does not exhibit an EC as high as that of graphene, graphene conductivity is 1 × 10^6^ S·m^−1^, and the rGO film conductivity measured in this experiment is only 15.51 S·m^−1^ (6.45 Ω·cm). The electrical resistivity (ER) of the samples were tested with a four-probe resistivity tester (Figure 7), the ER of the composites increased rapidly with decreasing rGO content. The thickest composite film prepared in this study was 26 μm and the thinnest was 21 μm; the effect caused by the small thickness difference was negligible. Figure 7 shows the calculated electrical conductivity of the samples. The EC increased gradually with increasing rGO content. Because of the many hydroxyl groups of NCC (which form intermolecular hydrogen bonds when blended with graphene in a manner that facilitates uniform intercalation into the middle of the graphene lamellae, and prevents graphene from π–π self-stacking), a more complete conductive network tends to form in the composite film with increasing graphene content [45].

The NCC/rGO sample exhibited an EC of 5.67 S·m^−1^, which is ca. 1000× higher than the ionic conductivity of the nanocellulose composite prepared by Willgert for use as an electrolyte in lithium-ion batteries (5 × 10^−3^ S·m^−1^) [46]. Table 2 shows the minimum resistivity and maximum conductivity of nanocellulose-based flexible conductive materials in recent years, and in comparison, NCC/rGO/PEG-1 has the best electrical properties. Therefore, composite films with excellent electrical conductivity can be prepared by combining NCC, GO and PEG. With 9.8 wt% added rGO, the EC of the composite film was 0.33 S·m^−1^. Because graphene exhibits properties—such as a large aspect ratio and specific surface area—that facilitate formation of conductive networks, the conductivity of the composite could also be substantially improved at smaller additions of rGO [47]. The synergistic effect between nanocellulose and graphene not only improves the dispersion properties of graphene, but also improves the orientation of the membrane matrix fibers, thus enhancing the electrical and mechanical properties of the composites [48].

### 3.6. Mechanical Property

Tensile strength of the samples were tested in accordance with GB/T 1040.3-2006 [59] executive standard. The toughness (Figure 8) was derived from the area integral enclosed by the stress–strain curve. The tensile strength of rGO/NCC was 16.71 MPa and the toughness was 313.86 kJ·m^−3^. After adding a small quantity of PEG, the tensile strength of the composite film decreased slightly, but the toughness increased to 482.93 kJ·m^−3^; which is because adding PEG decreased the number of hydrogen bonds between the NCC polymer chains and thus improved the stability of the film matrix [60]. With increasing PEG content the tensile strength continued to increase, but the toughness of the composite film slowly decreased because the increasing PEG content partially disrupted the original tight stacking of the lamellae in the composite film. PEG attaches to part of the NCC–rGO sheet layer and interacts with it, limiting the movement of the polymer molecular chains [61]. This is also consistent with the SEM and XRD results. When the PEG content was increased to 65.6 wt%, the tensile strength was 30.56 MPa, which is ca. 2.2× than the tensile strength (13.62 MPa) of the nanocellulose–graphene conductive film prepared with poly(lactic acid) by Liu [62]. Furthermore, the toughness of this sample reached a maximum of 727.18 kJ·m^−3^. Table 3 shows the tensile strengths of nanocellulose-based composite films in recent years, all of which are less than 31.68 MPa for NCC/rGO/PEG-2. Therefore, it indicates that the addition of PEG can improve the tensile strength of composite films. It can be seen from Figure 7 that the addition of PEG is also helpful to enhance the toughness of the composite film. Because a large quantity of PEG fills into the gap of the lamellae, adjacent lamellae are closely connected in a manner that is reminiscent of an adhesive filling the cracks, resulting in intermolecular forces that supplement the original weak bonding between PEG and NCC. At this time, the composite film contains more PEG with better ductility and water absorption compared with before adding PEG. Therefore, the mechanical properties of the composite film rGO/NCC/PEG-4 were optimal; the tensile strength of this sample was 1.83× higher than that of the rGO/NCC film without added PEG, and the toughness was 2.32× higher.

## 4. Conclusions

In this work, composite films were prepared for the first time by mixing NCC, rGO and PEG. TG demonstrated that the addition of rGO and PEG significantly improved the thermal stability of the composite films. Due to the better electrical conductivity of graphene, the NCC/rGO film with the highest graphene content (50 wt%) had the smallest resistivity of 17.82 Ω·cm. In addition, the NCC/rGO/PEG-4 with 65.6 wt% PEG content and only 21 μm thickness had a tensile strength of 30.56 MPa, which was 1.83 times than that of the control group (NCC/rGO film); the toughness was 727.18 kJ·m^−3^, which is 2.32 times than that of the control group. This confirms that the addition of PEG effectively improved the mechanical properties of the composite films, especially the elasticity. Overall, the preparation of NCC/rGO/PEG composite films presents new investigations in the field of nanocellulose material science which improves the development and application of nanocellulose in high performance composites. Meanwhile, since NCC and PEG are green and pollution-free, they can greatly reduce the burden of conventional conductive materials on the natural environment. The enhanced mechanical properties, especially flexibility, improve the application potential of conductive composite films in areas requiring high strength and toughness.

## Figures and Tables

**Figure 1 nanomaterials-12-04371-f001:**
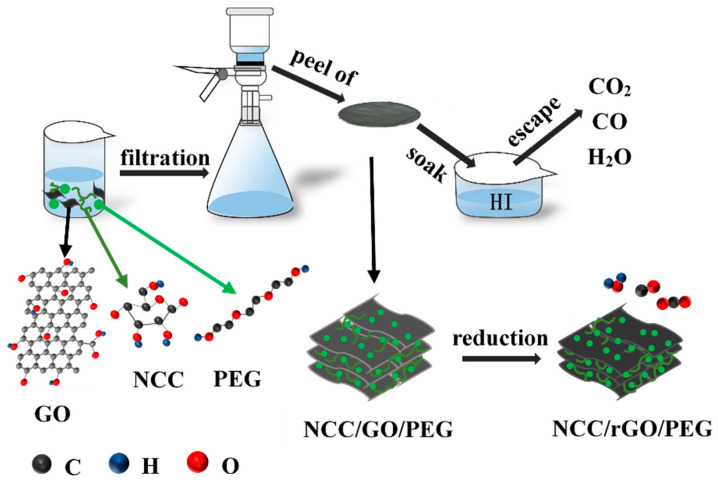
The main preparation process of the samples.

**Figure 2 nanomaterials-12-04371-f002:**
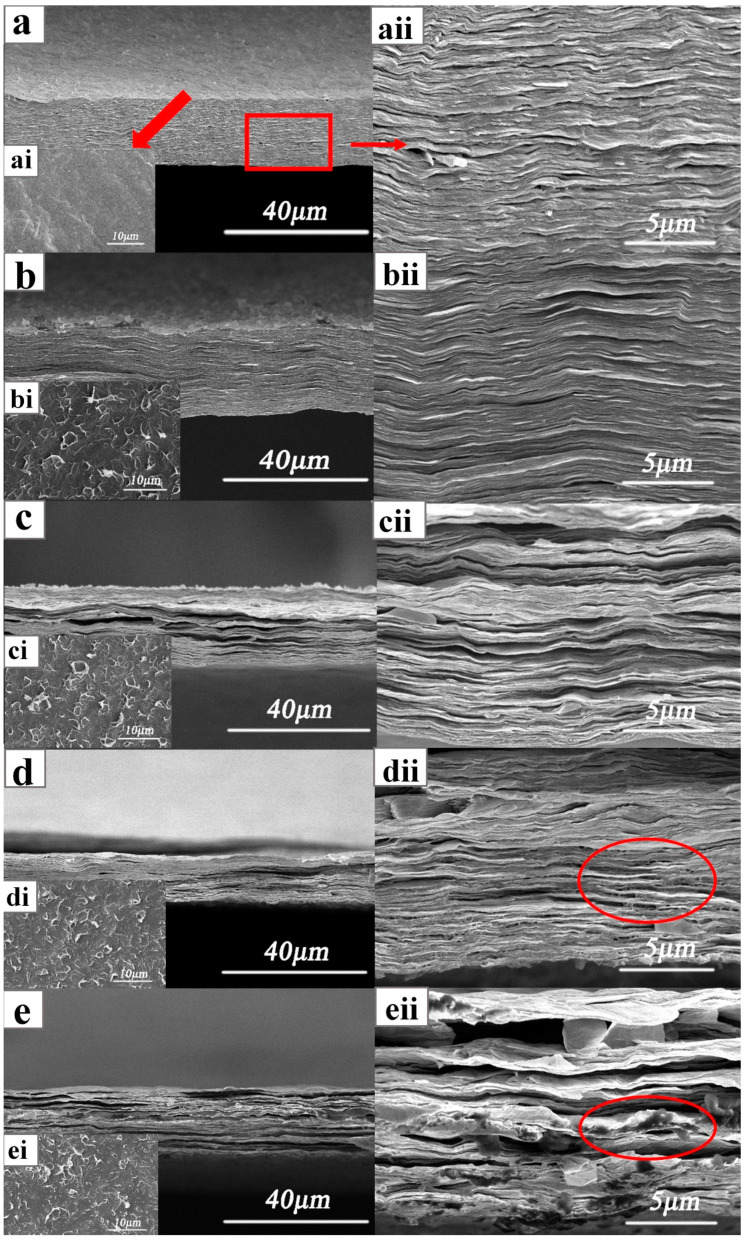
Cross-sectional and surface SEM images of composite films ((**a**–**e**) are images of the cross sections of NCC/rGO, NCC/rGO/PEG-(1–4), respectively; (**ai**–**ei**) are surface images of the samples; and (**aii**–**eii**) are enlarged cross-sectional regions).

**Figure 3 nanomaterials-12-04371-f003:**
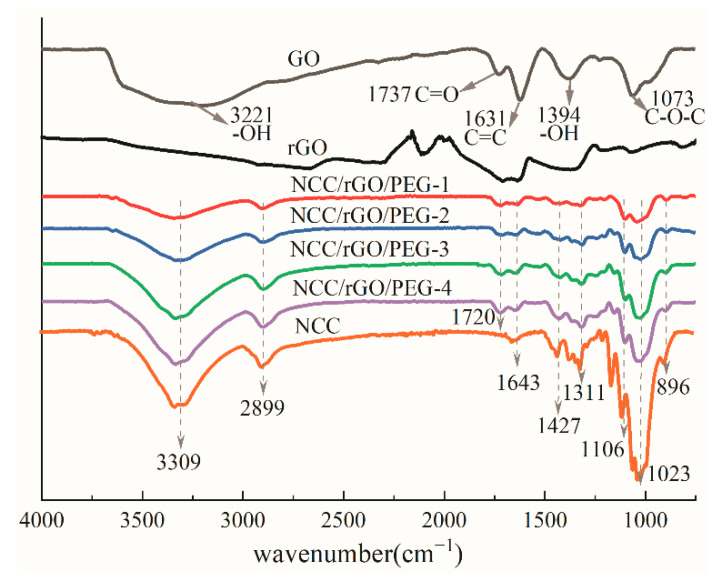
FTIR patterns of sample NCC, GO, rGO and different ratios of NCC/rGO/PEG composite films.

**Figure 4 nanomaterials-12-04371-f004:**
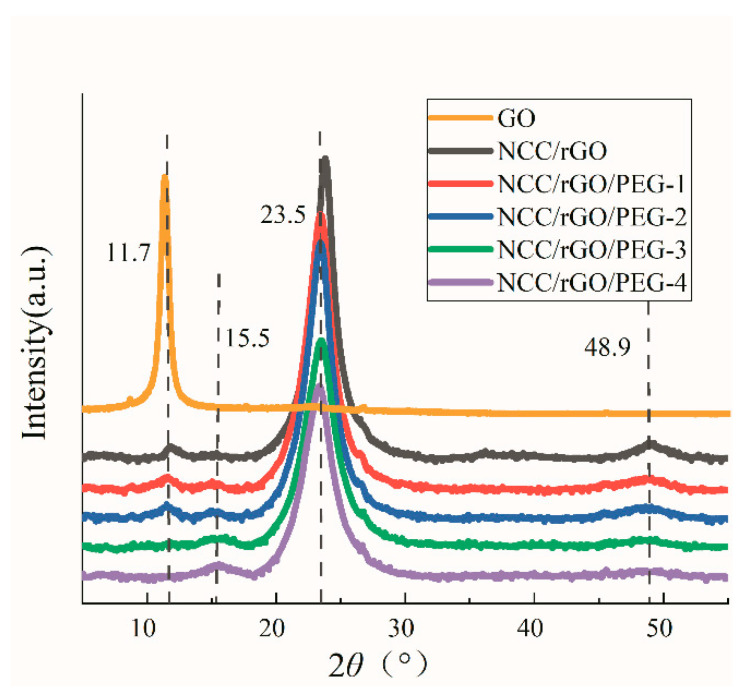
X-ray diffraction(XRD) spectrum of GO and the composite film.

**Figure 5 nanomaterials-12-04371-f005:**
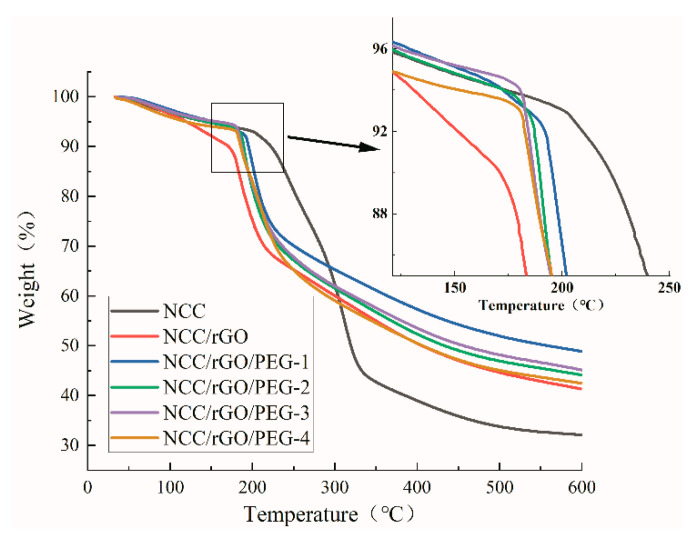
Thermalgravimetric analysis (TGA) of NCC film and its composite film.

**Figure 6 nanomaterials-12-04371-f006:**
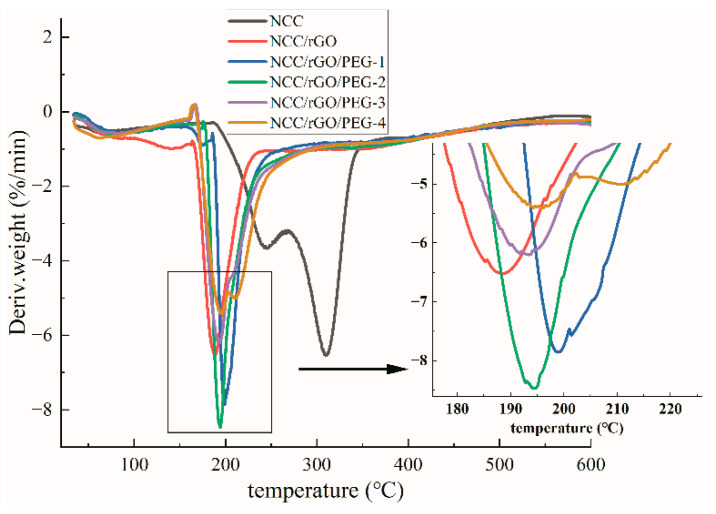
Derivative thermogravimetric (DTG) of NCC film and its composite film.

**Figure 7 nanomaterials-12-04371-f007:**
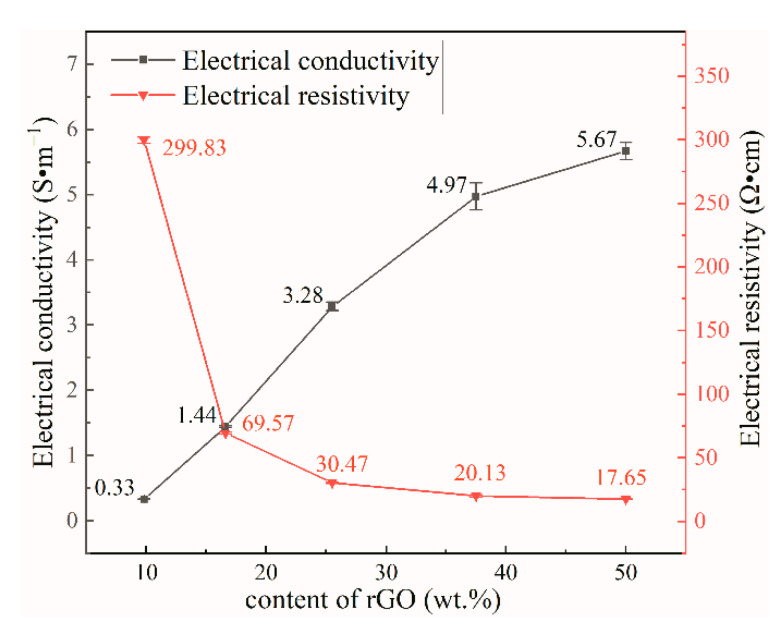
Electrical resistivity and conductivity of composite films with different rGO contents.

**Figure 8 nanomaterials-12-04371-f008:**
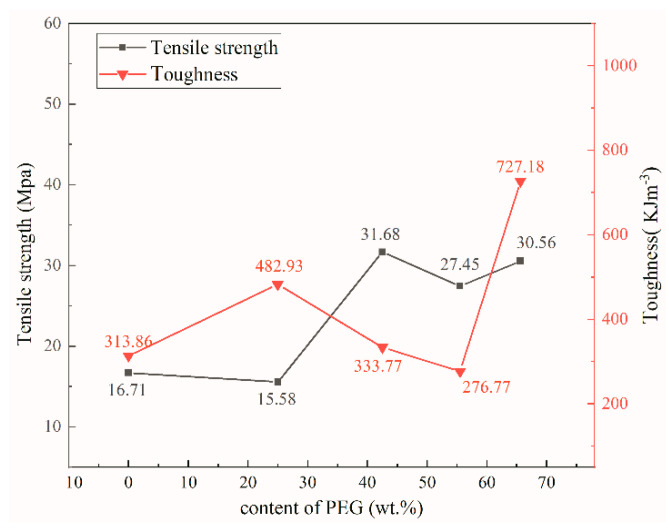
Tensile strength and toughness of composite films with different PEG contents.

**Table 1 nanomaterials-12-04371-t001:** Content of each component in composite films and the thickness of the composite film.

Composite Film	Thickness (μm)	NCC (wt.%)	rGO (wt.%)	PEG (wt.%)	NCC:GA:PEG
rGO	28	0	100	0	0:1:0
NCC/rGO	26	50	50	0	1:1:0
NCC/rGO/PEG-1	24	37.5	37.5	25	1.5:1.5:1
NCC/rGO/PEG-2	22	31.9	25.5	42.5	1.3:1:1.7
NCC/rGO/PEG-3	22	27.6	16.6	55.5	2.5:1.5:5
NCC/rGO/PEG-4	21	24.6	9.8	65.6	1.5:0.6:4.1

The initial content of PEG during the synthesis of samples.

**Table 2 nanomaterials-12-04371-t002:** Minimum resistivity and maximum conductivity of nanocellulose-based composite films.

Composite Film	Minimum ER (Ω·cm)	Maximum EC (S·m^−1^)	Reference
Graphite/regenerated cellulose film	8.3 × 10^3^		[49]
Cellulose-indium tin oxide layered film	2.5 × 10^3^		[50]
Nanocrystalline cellulose/modified multi-walled carbon nanotubes film	653.6		[51]
Graphene oxide/polyvinyl alcohol film	146		[52]
Electro-spinning oriented nanocellulose/carbon nanotubes/polyvinyl alcohol composite conductive film		0.12	[53]
Bacterial cellulose/halloysite nanotubes composite nanofiber film		0.513	[54]
Conductive cellulose nanofiber aerogels		3.72 × 10^−2^	[55]
Nanocellulosepolypyrrole composite		1. 5	[56]
Nanocellulose/polyaniline composite film		2.6 × 10^−3^	[57]
Graphene/cellulose composite fiber		0.141	[58]
Nanocrystalline cellulose- graphene- polyethylene glycol electrically film	20.13	4.97	This work

**Table 3 nanomaterials-12-04371-t003:** Maximum tensile strength of nanocellulose-based composite films.

Composite Film	Tensile Strength (Mpa)	Reference
nanocellulose-carboxylated carbon nanotube-graphite/polypyrrole flexible electrode composite	28.90	[63]
nanocrystalline cellulose/graphene flexible film	4.33	[64]
Cellulose Nanocrystal/Polyaniline/Dodecylbenzenesulfonic Acid film	22	[65]
Cellulose nanocrystal/polyaniline composite film	26.7	[66]
Chitosan/polyvinyl alcohol/cellulose nanocrystalline film	5.55	[67]
Sugar palm nanocellulose/sugar palm starch composite film	10.68	[68]
Nanocrystalline cellulose- graphene- polyethylene glycol electrically film	31.68	This work

## Data Availability

The data that support the findings of this study are available from the corresponding author upon reasonable request.

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
