# Peer review of "Mechanically Enhanced Nanocrystalline Cellulose/Reduced Graphene Oxide/Polyethylene Glycol Electrically Conductive Composite Film"

_nanomaterials, 2022, doi:10.3390/nano12244371_

Round 1
Reviewer 1 Report
The current paper deals with cellulose, GO and PEG blends for electronics applications. It is well-written and sound. While several shortcomings need to be solved before publishing:
1) Full details on the used equipment and characterization methods should be provided.
2) Section titles need to be in full names but not as acronyms.
3) Did use to measure the electrical conductivity of neat rGO?
4) The obtained results need to be compared with the literature data for similar/different systems and preparation methods. For example, cellulose paper preparation can be discussed in relation to the paper 10.1016/j.carpta.2022.100207. Many others also need to be used.
Author Response
Thank you for your valuable comments, please see the attachment for my response.

Reviewer 2 Report
The Manuscript “Mechanically enhanced nanocrystalline cellulose-graphene-polyethylene glycol electrically conductive composite film” by Pengbo Xie etc. is devoted to synthesis and study of flexible conductive materials based on rGO, PEG and NCC.
According to the obtained data, the authors use not reduced GO, but partially reduced GO. In this case, comparison in strength and other increase with graphene is impossible. However, this study shows how the properties of composites change depending on the ratio of components (GO, PEG and NCC). Also, the authors should replace the term "graphene" in the Title with "reduced graphene oxide"
In addition, below are some comments on the manuscript.
Row 77. It is necessarily to transcript PAM (polyacrilamide)
Row 81. GA, CNF are redundant abbreviations
Row 83. Mistaken “stabilityand”
Row 123. Authors need to indicate parameters of centrifugation (speed and time)
Row 146. HI was not listed in the Materials
Fig 1. Only ethylene glycol is imagined (only one monomer) on the Fig. 1, not PEG
Row 172. “With” need to write with small letter
Row 181. “The sample NCC/rGO/PEG-3 lamellae exhibited some dense mesh-like structures between the lamellas that acted as bridges connecting the adjacent lamellae [marked in Figure 2(cii)], because of the increase in the PEG content that facilitated interactions with the adjacent NCC between the lamellae and created intermolecular hydrogen bonds with GO. However, with increasing PEG content, there was a buildup [such as the dark gray block at the mark in Figure 2(dii)]; in which case PEG filled the gap between the lamellae and adhered to the lamellae in the manner of an adhesive, forming a new connection in the composite film”. It is very hard to read.
Row 142. Why was the mixture for the synthesis of composites stirred for a long time in an ice bath? Did the temperature rise after mixing of components?
Table 1. How was the content of PEG in composites investigated? Perhaps some of it could be removed from the composite after washing with water? Or does Table 1 contain only data on the initial content of PEG during the synthesis of samples?
SEM investigations. It would be interesting to compare the SEM images with a sample that does not contain PEG. In accordance with the marker (40 µm), sample 3 (Fig. 1c) has the smallest thickness, which is two times less than that of sample 1 (Fig. 1a).
Why did the authors use PEG with molar mass of 400, could it be better to prepare a composite with PEG 1500 and higher?
FTIR. The results of IR spectroscopy of reduced graphene oxide should be given. Also, the contribution of PEG, which is present in the composite, is not clear The authors talk about the incomplete reduction of graphene oxide, which is confirmed by the presence of C-O-C, but it is doubtful. An increase in the intensity of C-O-C vibrations is possible due to the IR performance conditions. Data of FTIR parameters (e.g. number of scans) are not listed in the relevant section. Film thickness is also important for IR spectroscopy. The parameters for XRD and SEM should also be supplemented.
XRD. It would be precise to give the XRD data for the synthesized graphene oxide. Maybe it is worth increasing the processing time of the samples when they are reduced with hydroiodic acid?
Author Response

(The authors gave the same response as above.)

Round 2
Reviewer 2 Report
The authors adequately responded to the comments. In the title of Manuscript, in accordance with the recommendation, the term "graphene" was changed to "reduced graphene oxide".